# Fangs in the Ghats: Preclinical Insights into the Medical Importance of Pit Vipers from the Western Ghats

**DOI:** 10.3390/ijms24119516

**Published:** 2023-05-30

**Authors:** Suyog Khochare, R. R. Senji Laxme, Priyanka Jaikumar, Navneet Kaur, Saurabh Attarde, Gerard Martin, Kartik Sunagar

**Affiliations:** 1Evolutionary Venomics Lab, Centre for Ecological Sciences, Indian Institute of Science, Bangalore 560012, India; suyogk@iisc.ac.in (S.K.); senjir@iisc.ac.in (R.R.S.L.); jpriyanka@iisc.ac.in (P.J.); navneetwg@gmail.com (N.K.); saurabh13attarde@gmail.com (S.A.); 2The Liana Trust, Survey #1418/1419, Rathnapuri, Hunsur 571189, India; gerry@gerrymartin.in

**Keywords:** *Hypnale hypnale*, *Craspedocephalus malabaricus*, *Craspedocephalus gramineus*, morbidity, preclinical research, antivenomics

## Abstract

The socioeconomic impact of snakebites in India is largely attributed to a subset of snake species commonly known as the ‘big four’. However, envenoming by a range of other clinically important yet neglected snakes, a.k.a. the ‘neglected many’, also adds to this burden. The current approach of treating bites from these snakes with the ‘big four’ polyvalent antivenom is ineffective. While the medical significance of various species of cobras, saw-scaled vipers, and kraits is well-established, the clinical impact of pit vipers from regions such as the Western Ghats, northeastern India, and the Andaman and Nicobar Islands remains poorly understood. Amongst the many species of snakes found in the Western Ghats, the hump-nosed (*Hypnale hypnale*), Malabar (*Craspedocephalus malabaricus*), and bamboo (*Craspedocephalus gramineus*) pit vipers can potentially inflict severe envenoming. To evaluate the severity of toxicity inflicted by these snakes, we characterised their venom composition, biochemical and pharmacological activities, and toxicity- and morbidity-inducing potentials, including their ability to damage kidneys. Our findings highlight the therapeutic inadequacies of the Indian and Sri Lankan polyvalent antivenoms in neutralising the local and systemic toxicity resulting from pit viper envenomings.

## 1. Introduction

The burden of death and disability due to snakebite in India is largely attributed to the ‘big four’ snakes: the spectacled cobra (*Naja naja*), Russell’s viper (*Daboia russelii*), common krait (*Bungarus caeruleus*) and saw-scaled viper (*Echis carinatus*) [1]. In reality, many other non-big four species significantly contribute to this socioeconomic disease [2]. While the medical importance of various (sub)species of cobras, saw-scaled vipers and kraits in India is very well-known [3,4,5,6,7,8,9,10,11], the clinical impact of many pit vipers remains underinvestigated [2]. Amongst the 90+ snake species found in the Western Ghats [12], only a few can potentially inflict clinically severe envenoming in humans. These include the hump-nosed viper (*Hypnale hypnale*), the Malabar pit viper (*Craspedocephalus malabaricus*), and the bamboo pit viper (*C. gramineus*).

The hump-nosed viper, in particular, has been documented to inflict coagulopathic and haemorrhagic complications in bite victims [13,14,15,16,17]. Envenomed victims have also been reported to exhibit symptoms such as swelling, retroperitoneal haemorrhage, thrombotic microangiopathy, regional lymphadenopathy, acute kidney injury and nephrotoxicity [18,19,20,21]. In addition, bite victims have also experienced acute ischemic stroke and electroencephalographic abnormalities [22,23]. In certain cases, the demise of the patients, as immediately as within the first thirty minutes of envenomation due to cardiac arrest, has also been reported in Sri Lanka [24]. Threat from *H. hypnale* is significant in certain regions of the Western Ghats, including the southern Indian state of Kerala, where the number of *Hypnale* envenomings outnumber those of the ‘big four’ snakes [25].

While the arboreal, camouflaged *C. malabaricus*, is known to be a major threat to cultivation and plantation workers in the Western Ghats [26,27], *C. gramineus* is not perceived as medically important. A recent report from a hospital in Karnataka suggested that the number of *C. malabaricus* envenomings surpassed those of the common krait and saw-scaled viper [28]. Envenoming from this species is known to induce myotoxicity, resulting in physical distortion, cellulitis, gangrene, haemotoxicity, hypotension and mild impairment of renal function [28,29]. Surprisingly, severe local tissue deterioration has also been recorded in bite victims [26]. Despite the irrefutable evidence provided by clinical reports, very few studies have attempted to comprehensively investigate the mechanisms of envenoming and the medical relevance of these snakes.

Given the dearth of knowledge, we compared the venom proteomic composition, biochemical and pharmacological activities and acute kidney injury (AKI) caused by *H. hypnale*, *C. malabaricus*, and *C. gramineus* from the Western Ghats. In general, the treatment strategies against pit viper bites involve the administration of the ‘big four’ Indian antivenoms. Therefore, to better understand the effectiveness of this treatment strategy, we preclinically evaluated the neutralising potencies of the big four Indian polyvalent antivenoms and a pentavalent Sri Lankan antivenom manufactured using a cocktail of immunogens that includes the venoms of the ‘big four’ snakes, as well as the venoms of *Hypnale* spp. from Sri Lanka. Our findings provide insights into the effectiveness of the current antivenom therapy against the pit vipers of the Indian Western Ghats.

## 2. Results

### 2.1. Venom Proteomics

The compositional variation in venoms of pit vipers from the Western Ghats was elucidated by sodium dodecyl sulphate polyacrylamide gel electrophoresis (SDS-PAGE) and tandem mass spectrometry. In-gel-digested SDS-PAGE bands were subjected to mass spectrometric analysis to determine the relative proteomic abundances of venom toxins. Significant venom variation was recorded between the three pit viper species, as evidenced by the SDS-PAGE band profiles and their intensities (Figure 1A). The most notable variations were seen between the 10–15 kDa and 20–75 kDa bands, especially in *H. hypnale* venom, where the intensities are significantly higher than in other pit viper venoms (Figure 1A).

The diverse compositional variation is also highlighted by mass spectrometric data. The venoms of *H. hypnale*, *C. gramineus* and *C. malabaricus* showed the presence of 45, 37, and 67 non-redundant protein families, respectively (Figure 1C; Appendix A). Toxin families, including snake venom serine protease (SVSP), snake venom metalloproteinase (SVMP), cysteine-rich secretory proteins (CRISP), phospholipase A_2_ (PLA_2_), vascular endothelial growth factor (VEGF), L-amino acid oxidase (LAAO), nerve growth factor (NGF), 5′-nucleotidase (5′-NTD), lectin, phosphodiesterase (PDE), phospholipase B (PLB), cystatin and hyaluronidase were identified in mass spectrometry experiments (Figure 1C; Appendix A).

Mass spectrometry data revealed that SVSP (44.5%) and CRISP (23.1%) toxins majorly constitute the venom proteome of *H. hypnale*, followed by lectins (17%) and LAAO (7.8%) (Figure 1C; Appendix A). While only 5% of the *H. hypnale* venom proteome was constituted by SVMP, it was identified as one of the major toxins in the venom proteomes of *C. gramineus* (21%) and *C. malabaricus* (12%) (Figure 1C; Appendix A). Similarly, to the venom proteome of *H. hypnale*, the venom of *C. malabaricus* was also dominated by SVSPs. The relative contribution of SVSP toxins in *C. malabaricus* and *C. gramineus* venoms was ~30% and 14%, respectively. Furthermore, nearly one-fourth of the *C. gramineus* and *C. malabaricus* venom proteome was constituted by PLA_2_ toxins (31% and 27.5%, respectively). CRISP was identified as the other major venom component in these snakes, with a relative abundance of 17.6% and 14.3% in *C. gramineus* and *C. malabaricus*, respectively.

### 2.2. Venom Biochemistry

The clinical implications of bites from pit viper species could be directly related to the functional diversity of their venom components. Since enzymatic components were found to constitute a major proportion of the three pit viper venom proteomes, in vitro assays were conducted to measure their relative activities. When tested for their ability to cleave a phospholipid (NOB) substrate, *C. gramineus* exhibited the highest PLA_2_ activity (*p* < 0.0001), followed by *C. malabaricus* and *H. hypnale* (Figure 2A).

Incidentally, the *C. gramineus* venom was also constituted by the highest amounts of PLA_2_ (30.6%; Figure 1C). While *C. gramineus* also exhibited the highest LAAO activity, the ability of *H. hypnale* to cleave L-amino acids was higher than that of *C. malabaricus* (Figure 2B). When tested against a protein (azocaesin) substrate, *H. hypnale* venom exhibited the highest total protease activity (*p* < 0.001), followed by *C. malabaricus* and *C. gramineus* (Figure 2C). It was also observed that the total protease activity was predominantly contributed by SVMP, as evidenced by the EDTA inhibition test group. However, the addition of SVSP inhibitors such as PMSF also reduced the overall activity, indicating synergy between the two toxin families. These differences in total protease activity are also supported by the combined abundances of SVMP and SVSP in the venoms (Figure 1C), wherein *H. hypnale* had the highest abundance of proteases (49.1%), followed by *C. malabaricus* (41.2%) and *C. gramineus* (34.7%).

When we assessed the fibrinogenolytic potential of *C. malabaricus* and *C. gramineus* venoms, we observed a complete degradation of Aα, Bβ and γ bands (Appendix A). The *H. hypnale* venom, however, cleaved Aα and Bβ bands only (Appendix A). While treatment of *C. gramineus* venom with EDTA, an SVMP inhibitor, did not inhibit fibrinogenolysis, treatment with PMSF, an SVSP inhibitor, prevented the cleavage of Bβ and γ bands but not Aα. Contrastingly, the cleavage of all three fibrinogen bands by *C. malabaricus* venom was inhibited in the presence of EDTA. However, in the case of *H. hypnale* venom, EDTA alone offers partial protection of the Bβ band, whereas PMSF does not inhibit fibrinogenolysis. A complete inhibition of fibrinogenolysis was observed for all three venoms in the presence of both SVMP and SVSP inhibitors.

### 2.3. Binding Efficiency and Cross-Reactivity of Commercial Antivenoms

The ability of commercial Indian and Sri Lankan antivenoms to bind to pit viper venoms was assessed using endpoint enzyme-linked immunosorbent assay (ELISA) experiments. Here, the antivenom was incubated with a fixed concentration of venom (100 ng), and the absorbance corresponding to the binding potential was recorded. While all four tested Indian antivenoms (i.e., Premium Serums, VINS, Virchow and Haffkine) exhibited a low titre of 1:100 against *C. gramineus*, the antivenom manufactured by Bharat Serums showed the least binding potential with a titre of 1:20 against this venom (Figure 3B). The Sri Lankan pentavalent antivenom manufactured by Premium Serums against five snake species, including *Hypnale* spp. from the island nation, showed a relatively better binding (1:500) against the Indian bamboo pit viper venom (Figure 3B). Additionally, the Sri Lankan antivenom and one of the Indian antivenoms (Virchow) exhibited relatively higher binding (1:2500) against *C. malabaricus* venom when compared to the other Indian antivenoms (1:500; Figure 3A). As the Sri Lankan antivenom was manufactured using an immunisation cocktail that included the venom of the Sri Lankan *Hypnale* species, its binding potential against the *H. hypnale* venom from the Indian Western Ghats was higher than the cross-recognition capabilities of the Indian ‘big four’ antivenom (a titre of 1:500 vs. 1:100; Figure 3C). Owing to relatively better binding potential, the Sri Lankan antivenom was downselected for the in vivo evaluation. Additionally, since all the Indian antivenoms performed similarly against pit viper venoms, only one of them was downselected as a representative for in vivo assays. Notably, since the Sri Lankan antivenom is also manufactured and marketed by Premium Serums, the Indian antivenom from the same manufacturer was shortlisted for comparative assessment.

### 2.4. Preclinical Assessments

#### Venom Toxicity

Toxicity profiles of pit viper venoms against mice were determined using WHO-approved assays. While a range-finding study with five individuals was carried out for *C. gramineus* and *C. malabaricus*, given the medical importance of *H. hypnale* in Sri Lanka and the Western Ghats, a complete LD_50_ experiment was performed for the latter using five distinct dose groups, each containing five mice (Appendix A). The LD_50_ of *H. hypnale* venom was estimated to be 1.32 mg/kg (or 26.48 μg/mouse). In contrast, the LD_50_ of *C. gramineus* (4 mg/kg) and *C. malabaricus* (6 mg/kg) venoms was three to five times higher, respectively (Table 1).

Preclinical experiments were also conducted to evaluate the morbidity-inflicting potentials of pit vipers from the Western Ghats. However, owing to the lack of clinical reports on morbid manifestations of *C. gramineus* bites, as well as animal ethics considerations, this species was excluded from these assessments. The outcomes of our experiments indicated a dose-dependent increase in the haemorrhagic potential of *C. malabaricus* and *H. hypnale* venoms (Figure 4A,D; Table 1). Interestingly, despite possessing a four-fold lower lethal potency in comparison to *H. hypnale*, the haemorrhagic potential of *C. malabaricus* venom (0.36 μg/mouse) was over six times higher than the former (2.3 μg/mouse; Figure 4A,D). In contrast, the MND of *H. hypnale* venom was estimated to be 27 μg in mice. As *C. malabaricus* venom did not exhibit necrosis, even up to 45 μg of venom in a range-finding study, complete experiments were not performed for this species.

### 2.5. Histopathological Evaluations

#### 2.5.1. Renal Histology

The nephrotoxic and myotoxic effects of pit viper venoms were assessed as outlined in the workflow (Figure 5).

Microscopic examination of kidney sections of mice that received an intravenous injection of venom corresponding to the half, one-fourth, and one-eighth dilutions of the LD_50_ of *C. malabaricus*, *C. gramineus*, and *H. hypnale* revealed glomerular degeneration associated with increased capsular space (Appendix A). The degeneration was documented in comparison to the control group that received normal saline (Appendix A). In addition, degenerative changes were also observed in the tubular regions, including loss of the proximal brush border and cytoplasmic vacuolation (Appendix A). When we consider Grade 4 glomerular damage, which represents the greatest damage to glomeruli, *H. hypnale* venom was documented to cause the most glomerular damage, irrespective of the route of venom injection: intravenous (Appendix A) or intradermal routes (Figure 6). Here, we see a consistent damage profile for the *H. hypnale* venom that increases with increasing concentrations, resulting in greater glomerular injury (G3 and G4). This was followed by *C. malabaricus* and *C. gramineus* venoms (Appendix A). In contrast, we observed the highest damage for *C. malabaricus* venom in TIS analysis, followed by *C. gramineus* and *H. hypnale* (Appendix A).

We also observe that *H. hypnale* venom results in glomerular degeneration at 15.4 µg and 19.25 µg venom doses, even when administered intradermally (Figure 6B,C and Figure 7A). Additionally, injections of 24.05 µg, 30.08 µg, and 37.06 µg of *H. hypnale* venom led to degenerative changes in the proximal tubule, including the loss of proximal brush boundary and cytoplasmic vacuolation (Figure 6D–F, and Figure 7B). Furthermore, tubular necrosis and sloughing of the tubule lining epithelium were observed in several kidney sections at 30.08 µg and 37.06 µg venom doses (Figure 6E,F and Figure 7B). Based on quantitative assessments of glomerular and tubular damage, grade 4 and grade 3 damage appear to increase with *Hypnale* venom concentration (Figure 7).

#### 2.5.2. Skeleton Muscle Histology

Myotoxicity assessments of pit viper venoms in this study revealed extensive muscle damage. Four test groups of three CD-1 mice were injected into their right gastrocnemius with venom concentrations corresponding to half, one-fourth and one-eighth of venom LD_50_. The histopathological observations of muscle tissue sections injected with venom showed degraded and disordered myofibrils (Appendix A) along with intramuscular haemorrhage (Appendix A). At a lower dilution of one-eighth LD_50_, muscle tissue injury was not observed (Appendix A).

#### 2.5.3. Neutralisation of Lethal and Morbid Effects of Pit Viper Venoms by Indian and Sri Lankan Polyvalent Antivenoms

Overall, in vitro binding experiments revealed poor cross-recognition of *H. hypnale*, *C. gramineus*, and *C. malabaricus* venoms by Indian and Sri Lankan antivenoms (Figure 3). However, since the pentavalent product from Sri Lanka (manufactured by Premium Serums against the Sri Lankan ‘big four’ snakes, as well as *Hypnale* spp.) was found to exhibit relatively better binding against the Indian *H. hypnale*, we downselected this antivenom for the in vivo experiments. Since all of the tested Indian products exhibited poor binding, we downselected the antivenom manufactured by Premium Serums for these experiments. Our findings revealed a complete preclinical failure of Indian and Sri Lankan antivenoms against the fatal effects of *H. hypnale* from the Western Ghats. Both products could not save the test population of animals when challenged with a 5× LD_50_ venom dosage. Furthermore, when the experiment was repeated against a lower challenge dose of 3× LD_50_, both antivenoms again failed to neutralise the lethal effects (Table 1; Appendix A).

As our preclinical evaluations highlighted considerable haemorrhagic and necrotising potentials of *H. hypnale* and *C. malabaricus* venoms, the neutralisation capabilities of the Indian and Sri Lankan antivenoms against these severities were tested. Although minimal amounts of Indian (3.6 × 10^−^^4^ mg/mL) and Sri Lankan (3.4 × 10^−^^4^ mg/mL) polyvalent antivenoms could reduce the *C. malabaricus* venom-induced haemorrhage by 50%, both antivenoms were ineffective against the *H. hypnale* venom-induced haemorrhage and necrosis, even in an undiluted state (i.e., one vial of the marketed product resuspended as per manufacturer’s protocol; Figure 4).

## 3. Discussion

### 3.1. Venom Compositions of Pit Vipers from the Western Ghats

The Western Ghats are one of India’s mega-diversity hotspots and an ecologically and geologically significant area for a diversity of endemic plants and animals, including several species of pit vipers. Despite being capable of inflicting potentially severe envenoming, the venoms of certain pit vipers from this region, namely *C. malabaricus*, *H. hypnale*, and *C. gramineus,* have been underinvestigated. While several studies have focused on characterising the Sri Lankan *H. hypnale* venoms and their clinical toxicology [14,15,19,20,30,31,32], the literature on pit viper venoms from the Western Ghats is limited to the cataloguing of constituent toxins and biochemical activities, with limited insights on clinical manifestations [33]. Mass spectrometric analysis of the venoms of the aforementioned pit viper species from the Western Ghats showed substantial variation in the relative abundances of toxin families. The venom toxin families identified in this study are consistent with previous reports [33,34].

Similarly to the other snake species, the published literature suggests the possibility of intraspecific variations in the composition of *H. hypnale* venoms. These variations could be attributed to the diversity of ecological and environmental factors, including prey availability and predator density across their geographic distribution [9,10,35,36]. Notably, certain individuals of Sri Lankan *H. hypnale* were found to express PLA_2_s and haemorrhagic SVMPs in high abundances [30], while the venoms of other individuals were enriched with PLA_2_s [32]. Interestingly, the venom proteome of *H. hypnale* from Karnataka Western Ghats assessed in this study was majorly constituted by SVSP (44.5%), while PLA_2_s and SVMPs were found in lower abundances (2–5%; Figure 1C). In contrast, the venoms of *C. gramineus* and *C. malabaricus* that share the range distribution with *H. hypnale* in the Western Ghats had high proportions of PLA_2_s, SVMPs and SVSPs. Consistently with their venom profile, the venoms of *C. gramineus* and *C. malabaricus* also exhibited the highest PLA_2_ activities in comparison to *H. hypnale* venom (Figure 2A). The proteolytic potential of pit vipers has been reported to be higher than that of the big four snakes [26]. However, unique patterns of proteolysis and fibrinogenolysis were observed here in this study. While *H. hypnale* exhibited the highest total proteolytic activity, the other two venoms exhibited the highest fibrinogenolytic activity, cleaving all three fibrinogen bands. Surprisingly, previous profiling of *C. malabaricus* venom reported the cleavage of Aα and Bβ fibrinogen bands [26], which contrasts with our findings, wherein all fibrinogen bands were cleaved (Appendix A). While the source of venom for the above study was not reported, these differential patterns could be attributed to the geographic variations in the abundances of SVMPs and SVSPs in *C. malabaricus* venoms. Another study also indicated that a significant proportion of *C. malabaricus* venom was constituted by LAAO [34], whereas the relative estimates from our study indicate that this toxin contributes to only 10% of the total venom proteome. Consistent with our venom proteome, the *C. malabaricus* venom also exhibited the least LAAO activity (Figure 2B). The non-enzymatic CRISPs, which were among the major components in all three pit viper species, were previously only reported to be in trace quantities [33]. These findings highlight that the toxin compositions and biochemical functions of pit viper venoms in the Western Ghats are as diverse and complex as the big four snake venoms [8,9,10,11].

### 3.2. Clinically Important Venoms of Pit Vipers from the Western Ghats

In the Indian subcontinent, snakebite mitigation and research have largely focused on the big four snakes, with a near complete disregard for many other medically relevant snake species, including pit vipers from the Western Ghats [2]. This is despite the fact that these neglected species can be much more medically important than the big four snakes in certain regions. For instance, *H. hypnale* was found to be responsible for one-fourth of bites in 1500 patients from a tertiary care centre in North Kerala, India. In Sri Lanka, the same species is the primary cause of snakebite cases [13]. Coagulopathies are documented as the primary symptom in *H. hypnale* envenoming, along with pain, swelling, haemorrhage, blistering, bruising, and regional lymphadenopathy [37]. Though the incidence of Malabar pit viper bites is lower, accidental envenomation induces enlargement of lymph nodes, local toxicity, hypotension and mild renal impairments [29]. Being an arboreal snake, *C. malabaricus* bites were predominantly reported around the head and face of the victims [28,29]. Such cases resulted in facial oedema, which could have potentially progressed to obstruct the airway or induce ophthalmia, compromising vision [28]. While *C. gramineus* envenomation rarely manifests any severe clinical symptoms, a peculiar case of bilateral occipital lobe infarction was observed in a bite victim from Coorg, Karnataka [38]. However, the pit viper venom toxins contributing to these medically significant symptoms are yet to be characterised.

Consistent with these reports, our preclinical assessments highlight the severe morbid symptoms caused by *H. hypnale* and *C. malabaricus* from the Western Ghats. Although only *H. hypnale* is considered to be medically important among the three species under investigation, some of the documented symptoms were relatively more severe in *C. malabaricus* than *H. hypnale*. For instance, the *C. malabaricus* venom was previously reported to be highly haemorrhagic, myotoxic and necrotic in comparison to *D. russelii* and *N. naja* venoms, despite being non-lethal to mice [26]. In line with this observation, the haemorrhagic activity of *C. malabaricus* venom tested here (MHD = 0.365 μg/mouse) was 6.5 times that of *H. hypnale* (MHD = 2.32 μg/mouse). Moreover, the haemorrhagic potential was found to be between two to ten times that of *E. carinatus* (MHD = 0.44 μg/mouse) and *D. russelii* (MHD = 2.19 μg/mouse), respectively [39]. Moreover, all three pit viper venoms were documented to be highly myotoxic (Appendix A).

The lethal potency and morbidity-inducing potentials of *H. hypnale* venoms vary across their range distribution [14]. For instance, the necrotising activity of *H. hypnale* from southern Sri Lanka (15 μg/mouse; [14]) was almost twice that of the *H. hypnale* venom analysed in this study (27 μg/mouse; Figure 3G). However, the haemorrhagic activity of this venom (3.4 μg/mouse, [14]) was 1.5 times less than that of the *Hypnale* from the Western Ghats (2.32 μg/mouse; Figure 3D). In contrast, *H. hypnale* venom sampled from southwestern Sri Lanka exhibited reduced haemorrhagic (10.5 μg/mouse) and necrotising (39.3 μg/mouse) effects [40], compared with the Western Ghats and the other Sri Lankan populations [41]. As variations in the mortality- and morbidity-inducing potentials of venoms determine the severity of *H. hypnale* bites, further research is warranted to identify medically relevant populations across their range distribution.

### 3.3. Snakebite-Associated Acute Kidney Injury (SAKI)

SAKI is a poorly understood complication that has been linked to substantial morbidity and mortality. In clinical situations, the onset of SAKI can range from a few hours to a month after the bite [42]. The reported incidence of SAKI globally ranges from 8 to 60%, among which 15 to 92% of cases require some form of kidney replacement therapy, with an overall mortality of up to 45% [43]. Alarmingly, up to 50% of SAKI patients progress to develop chronic kidney disease [44]. Enzymatic toxins in snake venom result in injuries to all kidney cell types, including glomerular, tubulointerstitial, and kidney vasculature [43]. Pathogenesis includes ischaemia, decreased renal blood flow, proteolytic degradation of the glomerular basement membrane, thrombotic microangiopathy, cytotoxic, rhabdomyolysis, and the accumulation of large amounts of myoglobin in kidney tubules [45,46]. The pathophysiology of AKI in snake envenoming is unknown, and there is little evidence of primary nephrotoxins in snake venoms. We attempted to document structural alterations to the kidneys caused by the three pit viper venoms: *C. malabaricus*, *C. gramineus*, and *H. hypnale*. We observed that *H. hypnale* venom causes the highest grades of glomerular damage (i.e., grades 4 and 3), regardless of the route of venom injection: intravenous or intradermal. In intradermal injections, we observed that *H. hypnale* venom results in glomerular degeneration even at the lowest venom concentrations of 15.4 µg (19%) and 19.25 µg (Figure 6 and Figure 7). In the intravenous route of injection, we see a consistent increase in the glomerular damage profile with increasing venom concentrations for *H. hypnale*, followed by *C. malabaricus* and *C. gramineus* (Appendix A). Surprisingly, we observed the highest damage for *C. malabaricus* venom in TIS analysis, followed by *C. gramineus* and *H. hypnale* (Appendix A). These findings suggest that the three pit vipers from the Western Ghats target distinct parts of the nephron. Overall, these results emphasise the need to investigate the nephrotoxic potential of snake venom, as they may often go unnoticed in clinical cases.

### 3.4. Preclinical Failure of Commercial Polyvalent Antivenoms Highlights the Urgent Need for Region-Specific Antivenom Therapy

Preclinical studies have reported an effective cross-neutralisation of Sri Lankan *H. hypnale* venoms with non-specific monovalent (*Calloselasma rhodostoma*) and polyvalent (*C. albolabris*, *D. siamensis*, and *C. rhodostoma*) antivenoms [40]. Indian polyvalent antivenoms, raised against the big four snakes, are frequently administered for the treatment of pit viper envenoming. However, observations have shown that this strategy offers limited to no help [13]. The infusion of large quantities of non-specific antivenom products could result in unwanted secondary effects [47]. Nonetheless, given the inability of most clinicians to accurately identify snake species, a large number of antivenom vials (up to 40) are often administered to the bite victims of *H. hypnale*, *C. malabaricus* and *C. gramineus* (KS’ personal communication with clinicians). In line with the clinical reports, our findings reveal a preclinical failure of the ‘big four’ Premium Serums antivenom in protecting mice injected with the *H. hypnale* venom, even when the challenge dose is reduced from 5× to 3× (Table 1). Furthermore, we tested the usefulness of the pentavalent Sri Lankan antivenom, which includes *H. hypnale* venom from the island nation, in countering toxicities caused by *H. hypnale* from the Indian Western Ghats. Similarly to its Indian antivenom counterpart, Sri Lankan pentavalent product failed to neutralise the lethal effects of the venom, pointing to the geographical variation in *H. hypnale* venoms across the Indian subcontinent.

## 4. Materials and Methods

### 4.1. Venom and Antivenom Samples

Venoms of *H. hypnale*, *C. malabaricus* and *C. gramineus* were collected from various locations in the Western Ghats across the states of Maharashtra, Karnataka, and Kerala with due approval from respective forest departments (Maharashtra: (Desk-22 (8)/WL/Research/CR-60 (17–18)/2708/2018–2019)), Karnataka: (PCCF(WL)/C1(C3)/CR-09/2017–18), and Kerala (WL12-5395/2009:05012010). All venoms investigated were collected from adult specimens. However, given the forest department restrictions, invasive gender assessments were not performed. Extracted venom was flash-frozen, lyophilised and stored at −80 °C until further use. Detailed sampling information regarding locations, the number of individuals, as well as antivenoms investigated in this study are provided in Appendix A.

### 4.2. Proteomic Analyses

#### 4.2.1. Protein Estimation, One-Dimensional Gel Electrophoresis and In-Gel Digestion

The bovine serum albumin (BSA, Sigma-Aldrich, St. Louis, MO, USA) standard was used in a Bradford assay to determine the protein concentration of pit viper venoms [48]. Variations in the proteomic profiles of pit viper venoms were determined using sodium dodecyl sulphate polyacrylamide gel electrophoresis (SDS-PAGE). In brief, 20 µg venom samples were reduced using β-mercaptoethanol (Sisco Research Laboratories Pvt. Ltd., Mumbai, India) and subjected to 12.5% SDS-PAGE along with a protein molecular weight ladder [Precision Plus Dual Color (Bio-Rad Laboratories, New York, NY, USA)] at a constant voltage of 80 V [49]. Following electrophoresis, the gel was stained overnight with Coomassie Brilliant Blue R-250 (Sisco Research Laboratories Pvt. Ltd., Mumbai, India), destained and then imaged using an iBright CL1000 gel documentation system (Thermo Fisher Scientific, Waltham, MA, USA). Densitometric analysis of each band was performed using the ImageJ software, version 1.53t [50].

Individual venom bands were excised and subjected to in-gel digestion, followed by tandem mass spectrometry. Briefly, the excised gel bands were destained and then dehydrated with 50% acetonitrile. Following dehydration, the samples were reduced by treatment with 10 mM dithiothreitol (DTT) at 56 °C for 1 h. This was followed by a 30 mM iodoacetamide treatment for alkylation at room temperature for 45 min under dark conditions. A mixture of 25 mM ammonium bicarbonate in water and acetonitrile solution (1:1, *v*/*v*) was used to wash the bands. The excess solvent was removed using a vacuum concentrator (Thermo Fisher Scientific, Waltham, MA, USA). Samples were then digested with trypsin (0.2 µg/µL) overnight at 37 °C, and the peptides were extracted the next day into 50 μL of 50% acetonitrile solution.

#### 4.2.2. Liquid Chromatography-Tandem Mass Spectrometry (LC-MS/MS)

The relative abundance of various toxins present in pit viper venoms was determined by subjecting the individually excised venom bands to tandem mass spectrometry [51]. Briefly, the individual bands were reduced, alkylated, trypsinised and injected into the liquid chromatographic system programmed at a constant flow rate of 300 nL/min for a total run time of 120 min with varying concentrations of buffer A (0.1% formic acid in HPLC grade water) and buffer B (0.1% formic acid in 80% acetonitrile). The buffer gradient was set as 10–45% over 98 min, 45–95% over 4 min and 95% over 18 min. The eluted fractions were then introduced into a Thermo Orbitrap Fusion Mass Spectrometer (Thermo Fisher Scientific, Waltham, MA, USA) for tandem mass spectrometry. The MS scans were performed with the following parameters: range (*m*/*z*) of 375–1700 with a resolution of 120,000 and a maximum injection time of 50 ms. Furthermore, an ion trap detector with high collision energy fragmentation (30%) was used to perform fragment scans (MS/MS) with a range (*m*/*z*) of 100–2000 and a maximum injection time of 35 ms.

The identities of the individual toxins in each of the excised bands were determined by searching the raw MS/MS spectra against the NCBI-NR Serpentes database (taxid: 8570) using PEAKS Studio X Plus (Bioinformatics Solutions Inc., Waterloo, ON, Canada). The following parameters were defined for the search: the parent and fragment mass error tolerance limits were set to 10 ppm and 0.6 Da, respectively. A ‘monoisotopic’ precursor ion search type with ‘semispecific’ trypsin digestion, allowing for a maximum of three missed cleavages, cysteine carbamidomethylation (+57.02) as a fixed modification and methionine oxidation (+15.99) as a variable modification. The filtering parameters for match acceptance were set to a −10 lgP protein score of ≥20, detection of 1 unique peptide, and a false discovery rate (FDR) of 0.1%. The raw mass spectrometry data have been made available at the ProteomeXchange Consortium via the PRIDE partner repository [52] with data identifier: PXD039139 (Reviewer account details: Username: reviewer_pxd039139@ebi.ac.uk; Password: IvElDmOs). Hits with at least one unique matching peptide were considered for the downstream analyses, and the redundant protein hits from each protein family were removed manually. The relative abundance of each toxin hit in a fraction was determined by estimating the area under the spectral curve (AUC). However, these AUC values obtained from PEAKS Studio analyses, representing the mean spectral intensities, were, in turn, normalised across the gel bands using the densitometric estimates from the SDS-PAGE profiles [53]. The relative abundance of a protein family hit (X) was estimated using the equation below, where ‘N’ indicates the number of bands in the SDS-PAGE profile.
$$Relative\;abundance\;of\;X\;(\%)\;=\;\sum_{n=1}^{N}\frac{AUC\;of\;X\;in\;band\;Bn\;\times \;Density\;of\;Band\;B\;(\%)}{Total\;AUC\;if\;all\;protein\;families\;in\;the\;band\;nFn}$$

### 4.3. Biochemical Characterization

#### 4.3.1. Colorimetric Phospholipase A_2_ (PLA_2_) Assay

A colorimetric method was used to determine the phospholipase activity of venom, wherein a chromogenic lipid substrate, 4-nitro-3-[octanoyl oxy] benzoic acid (NOB; Enzo Life Sciences, New York, NY, USA), was used [54,55]. Briefly, 5 µg of the venom was added to 500 mM NOB substrate, dissolved in a 200 µL reaction buffer (10 mM Tris–HCl, 10 mM CaCl_2_, 100 mM NaCl; pH 7.8). The reaction was carried out at 37 °C for 40 min, and the absorbance was measured at 425 nm every 10 min. Thereafter, a standard curve with varying concentrations of the NOB substrate (4 nanomoles to 130 nanomoles) and 4 M NaOH was plotted, and the amount of the phospholipid substrate (nmol) cleaved per minute per mg of the venom was calculated.

#### 4.3.2. L-amino Acid Oxidase (LAAO) Assay

Pit viper venoms were tested for their LAAO activity following a previously described protocol [56]. The reaction was set up in triplicates, where 10 μg of the snake venom was added to L-leucine substrate solution [Tris-HCl buffer (50 mM), L-leucine (5 mM), horseradish peroxidase (5 IU/mL), and o-phenylenediamine dihydrochloride (2 mM)] and incubated at 37 °C for 60 min. Post this, the reaction was stopped by adding 2 M H_2_SO_4_, and the absorbance was measured at 492 nm with an Epoch 2 microplate spectrophotometer (BioTek Instruments, Inc., New York, NY, USA).

#### 4.3.3. Snake Venom Protease Assay

Pit viper venoms were assessed in a protease assay using a previously established protocol [57], where 10 µg of the venom was incubated with the azocasein substrate (400 µg) at 37 °C in triplicates for 90 min. This reaction was stopped by the addition of 200 µL trichloroacetic acid and subjected to centrifugation at 1000× *g* for 5 min. Thereafter, the supernatant was collected and mixed with equal volumes of 0.5 M NaOH. The absorbance was measured in an Epoch 2 microplate spectrophotometer (BioTek Instruments, Inc., New York, NY, USA) at 440 nm. To calculate the relative protease activity, a purified protease from the bovine pancreas was used as a positive control. Additionally, to understand the relative contribution of snake venom metalloproteinase (SVMP) and snake venom serine protease (SVSP) to the overall protease activity, pit viper venoms were incubated with 0.1 M ethylenediaminetetraacetic acid (EDTA) and/or 0.04 M phenylmethylsulfonyl fluoride (PMSF), respectively, at 37 °C for 15 min prior to the addition of azocasein.

#### 4.3.4. Fibrinogenolytic Assay

The ability of pit viper venoms to degrade the human fibrinogen was evaluated by utilising the method described previously [51,58]. Pit viper venoms (1.5 µg each) were incubated with 15 µg of human fibrinogen (Sigma-Aldrich, St. Louis, MO, USA) dissolved in phosphate buffer saline (PBS) at 37 °C for 60 min. A control with only PBS and human fibrinogen was also included. Post incubation, an equal volume of the loading dye (1 M Tris-HCl pH 6.8, 50% glycerol, 0.5% bromophenol blue, 10% SDS, 20% β-mercaptoethanol) was added, and the reaction mixture was heated at 70 °C for 10 min and subjected to 15% SDS-PAGE. The gel was stained with Coomassie Brilliant Blue R-250 and destained before visualisation in an iBright CL1000 (Thermo Fisher Scientific, Waltham, MA, USA) gel documentation system. A similar experiment was conducted in the presence of inhibitors, wherein the pit viper venoms were incubated with 0.1 M EDTA (SVMP inhibitor) and/or 0.04 M PMSF (SVSP inhibitor) at 37 °C for an additional 15 min prior to mixing with human fibrinogen [59].

### 4.4. Enzyme-Linked Immunosorbent Assay (ELISA)

In vitro venom recognition potential of commercially available antivenoms against pit viper venoms was assessed by performing indirect ELISA experiments [2]. First, 96-well microplates were coated with venom samples (100 ng) diluted in a carbonate buffer (pH 9.6) and incubated overnight. The unbound venom was washed off with Tris-buffer saline containing Tween 20 (0.01 M Tris pH 8.5, 0.15 M NaCl, 1% Tween 20), and the plates were incubated with blocking buffer (5% skimmed milk in TBST) for 3 h at room temperature. Serially diluted commercial Indian antivenoms (Premium Serums, VINS, Bharat Serums, Haffkine and Virchow) were added at the end of three-hour incubation with an intermediate round of TBST washing. In addition to the Indian antivenoms, the Sri Lankan pentavalent antivenom manufactured by Premium Serums was also examined. Plates were incubated overnight, and another round of washing with TBST was performed the next day. Thereafter, they were incubated at room temperature for 2 h following the addition of horseradish peroxidase (HRP)-conjugated, rabbit anti-horse secondary antibody (Sigma-Aldrich, St. Louis, MO, USA), diluted at a ratio of 1:1000 in PBS. Eventually, 100 μL of ABTS (2,2′-azino-bis (3-ethylbenzothiazoline-6-sulfonic acid); Sigma-Aldrich, St. Louis, MO, USA) substrate solution was carefully added to each well, and the absorbance was measured at a wavelength of 405 nm for 40 min in Epoch 2 microplate reader. A naive horse IgG (Bio-rad, Hercules, CA, USA) was used to determine the cut-off for non-specific antibody binding. The cut-off value, which is calculated as the mean absorbance of the naïve horse IgG plus two times the standard deviation, was considered for calculating the titres. The titre is the first dilution of the antivenom at which the mean absorbance value of the test samples is above the cut-off value [2,60].

### 4.5. Preclinical Assessments

#### 4.5.1. The Median Lethal Dose (LD_50_)

The median lethal dose or LD_50_ of pit viper venoms was evaluated in a murine model of envenoming. Five concentrations of each venom were diluted in physiological saline (0.9% NaCl), and 200 μL of this mixture was injected intravenously into the caudal vein of mice. Each venom dose group contained five male CD-1 mice (18–22 g). Additionally, 200 μL of normal saline was injected into a single mouse as a negative control. Following a 24-h observation period, the death–survival pattern was recorded to calculate the LD_50_ of the respective venom using probit statistics [61]. Owing to the significant medical relevance of *H. hypnale*, we only performed a complete LD_50_ experiment for this species. On the other hand, only dose-finding experiments involving one mouse per dose group were performed for *C. malabaricus* and *C. gramineus* venoms.

#### 4.5.2. The Median Effective Dose (ED_50_)

The neutralising potency of the ‘big four’ Indian and *Hypnale*-specific Sri Lankan antivenoms (i.e., the amount of venom in mg neutralised per ml of the reconstituted antivenom) was evaluated in the mouse model of envenoming. Given ethical considerations, only a single Indian antivenom with relatively better in vitro binding capability was tested. Various dilutions of Indian or Sri Lankan antivenom were incubated with a ‘challenge dose’ of *H. hypnale* venom (corresponding to 3× or 5× LD_50_) at 37 °C for 30 min. Following incubation, four groups of mice, with five individuals per group, were administered with these mixtures through an intravenous caudal vein injection. The experimental animals were maintained under observation for 24 h post-injection, and the death and survival patterns were recorded.

#### 4.5.3. The Minimum Haemorrhagic Dose (MHD)

The haemorrhagic abilities of pit viper venoms were evaluated by determining the minimum haemorrhagic dose (MHD) or the amount of venom in μg that induces a 10 mm haemorrhagic lesion within three hours of intradermal venom injection in mice [62]. Five graded concentrations of venoms were prepared in physiological saline (50 µL) and administered intradermally into five groups containing five mice each. Mice were humanely euthanised three hours post venom injection, and the diameter of the haemorrhagic lesion on the dorsal skin patch was measured using a vernier calliper. As a negative control, 50 µL of physiological saline was administered to a fifth group of mice.

#### 4.5.4. The Minimum Necrotic Dose (MND)

The minimum necrotising dose of the venom (MND) is defined as the amount of venom in μg that induces a 5 mm necrotic lesion in mice within 72 h of intradermal venom injection [63]. MND for the pit viper venoms was determined by intradermally injecting five graded venom concentrations diluted in physiological saline (50 µL) into five CD1 male mice (18–22 g). After 72 h of venom injection, mice were humanely euthanized, and the dorsal skin patch was examined for necrotic lesions. The fifth group of mice, where only the physiological saline was administered, served as the negative control.

### 4.6. Neutralisation of Venom-Induced Morbidities

The neutralisation potential of Indian and Sri Lankan polyvalent antivenoms against *H. hypnale* and *C. malabaricus* venom-induced morbidity was also evaluated. The median effective dose (MHD-ED_50_) of antivenom, which is defined as the minimum amount of antivenom in µL required to reduce the diameter of the haemorrhagic lesion by 50%, three hours post-injection of the venom challenge dose (5× venom MHD), was assessed in mice [64]. Five dilutions of the best-binding Indian antivenom or the Sri Lankan pentavalent antivenom were incubated with the challenge dose for 30 min at 37 °C. Post incubation, five dose groups of CD-1 male mice were intradermally injected with the mixture and monitored for three hours. The animals were then humanely euthanized by CO_2_ asphyxiation, and the diameter of the lesions was measured using a vernier calliper [64].

Similarly, the neutralising potency of antivenoms against the necrotising effects of *H. hypnale* venom was also evaluated. Here, 2× MND of venom was chosen to challenge the Indian and Sri Lankan antivenoms. MND_50_ of the antivenom was determined by quantifying the minimum amount of antivenom in µL required to diminish the diameter of the necrotic lesion by 50%. Test animals were humanely euthanized following a 72-h observation period, dissected, and the diameter of the necrotic lesion was measured [64].

### 4.7. Histopathological Evaluations

#### 4.7.1. Renal Histology

We evaluated the nephrotoxic potential of pit viper venoms by injecting concentrations equivalent to half, one-fourth, and one-eighth of LD_50_ in triplicates into the caudal vein of CD-1 male mice. Following a 24-h observation period, mice were humanely euthanized, and their kidneys were harvested. The nephrotoxic effects of *H. hypnale* venom were examined by evaluating kidneys harvested from the MND test population of mice that had received venom doses ranging from 19.04 μg to 37.06 μg via the intradermal route. Kidneys harvested after 72 h of the observation period were washed in 1X PBS, fixed in 10% buffered formalin for 24 h, and dehydrated with ascending concentrations of ethyl alcohol (70% and 95% for 15 min; 100% for 1 h 45 min), and cleared in xylene (Thermofisher, Waltham, MA, USA). Tissues were embedded in paraffin (Thermofisher, Waltham, MA, USA) at 58 °C, following which 3 μm sections were prepared using a Leica microtome (RM2245, Wetzlar, Germany). The slides were stained with haematoxylin (Leica, Wetzlar, Germany) and eosin (Leica, Wetzlar, Germany). Further, the slides obtained were visualised using an Olympus light microscope (Ix81, Shinjuku, Japan) at a 40× magnification, and images were acquired and analysed using CellSens dimension imaging software Ver 3.1.1 (Olympus, Shinjuku, Japan). The histological structure of renal tubules and glomeruli of the treatment group was compared to the control group, which received 50–200 μL of normal saline.

For scoring glomerular and tubular injury severity, stained sections of 60 subcapsular and 60 juxtamedullary regions from the control and experimental groups with five mice per venom concentration were examined. The severity was graded from 1 to 4 based on the area proportion of glomerular/tubular region involvement and graded according to the formula:$$GIS\;=\;\frac{\left(1\;\times \;G2\right)\;+\;(2\;\times \;G3)\;+\;(3\;\times \;G3)}{GTotal}\;\times \;100$$

Here, G2, G3, and G4 refer to the number of grade 2, grade 3, and grade 4 glomeruli, respectively, and G_Total_ refers to the total number of glomeruli counted.

The glomerular damage was assessed using the criteria of the collapse of the capillary lumen, glomerular basement membrane folding, and dark profiles in a glomerular tuft. A grade 2 injury involves up to 25% of the glomerulus, whereas a grade 4 injury involves more than 50% of the renal region. To calculate the glomerular injury score (GIS), we multiplied the number of glomeruli with a value of 1 by 0, 2 by 1, 3 by 2, and 4 by 3. These numbers were summed and divided by the number of glomeruli examined, including those with a score of zero [65].
$$TIS\;=\;\frac{\left(1\;\times \;T2\right)\;+\;\left(2\;\times \;T3\right)\;+\;\left(3\;\times \;T3\right)}{TTotal}\;\times \;100$$

Here, T2, T3, and T4 refer to the number of grade 2, grade 3, and grade 4 tubular injury, respectively, and T_Total_ refers to the total number of tubular sections counted.

The tubular injury was defined as tubular dilation, tubular atrophy, tubular cast formation, sloughing of tubular epithelial cells, loss of the brush border, and/or thickening of the tubular basement membrane. We used the following scoring system: 1 and 4 as a percentage of the total cortical area (1 = none, 2 ≥ 1–25%, 3 ≥ 25–50%, and 4 ≥ 50%). To calculate the tubular injury score (TIS), we multiplied the number with a value of 1 by 0, 2 by 1, 3 by 2, and 4 by 3. These numbers were summed and divided by the number of tubules examined, including those with a score of zero [66].

#### 4.7.2. Skeletal Muscle Histology

Four test groups of three CD-1 mice each were injected into their right gastrocnemius muscle with pit viper venom concentrations corresponding to half, one-fourth, and one-eighth of the respective LD_50_. Control mice were injected with saline alone. After three hours of injection, mice were sacrificed, and samples of the injected muscles were resected and fixed in 10% buffered formalin for 48 h. All fixed tissues were washed slowly with running tap water for a minimum of 30 min, decalcified with a working solution containing 8% hydrochloric acid and 8% formic acid for 24 h, and neutralised with ammonia solution for 30 min. All specimens were washed again in slowly running tap water thoroughly for up to 24 h, dehydrated with ascending concentrations of ethyl alcohol (70 and 95% for 1 h; 100% for 3 h) and cleared in xylene (Thermofisher, Waltham, MA, USA), followed by paraffin embedding. Then, 5 μm thick sections were obtained from each sample and stained with Masson’s Trichrome stain (MTS; Path Stains, India). Slides obtained were visualised using an Olympus light microscope (Ix81, Shinjuku, Japan) at a 20× magnification, and images were acquired and analysed using CellSens dimension imaging software Ver.3.1.1 (Olympus, Shinjuku, Japan). The histological structure of the skeletal muscle in the treatment group was compared to the control that received 50 μL of normal saline.

### 4.8. Statistical Analysis

Statistical comparisons between samples were carried out using an unpaired *t*-test in GraphPad Prism (GraphPad Software 9.0, San Diego, CA, USA, www.graphpad.com, accessed on 15 February 2023).

## 5. Conclusions

Proteomic and biochemical analyses of *H. hypnale*, *C. gramineus* and *C. malabaricus* venoms revealed considerable differences in composition and function. In support of clinical reports, our histopathological analyses revealed significant levels of injury to kidneys, highlighting the nephrotoxic potential of these snakes. Moreover, the venoms of *C. malabaricus* and *H. hypnale* were found to exhibit significant haemorrhagic and necrotising effects, respectively. Furthermore, preclinical experiments show the inadequacies of both Indian and Sri Lankan antivenoms in neutralising venom-induced mortality and morbidities. The morbidity-inducing potential of pit viper venoms, and the preclinical failure of Indian and Sri Lankan antivenoms in neutralising them, highlights the urgent need for region-specific antivenom therapy in the Indian subcontinent.

## Figures and Tables

**Figure 1 ijms-24-09516-f001:**
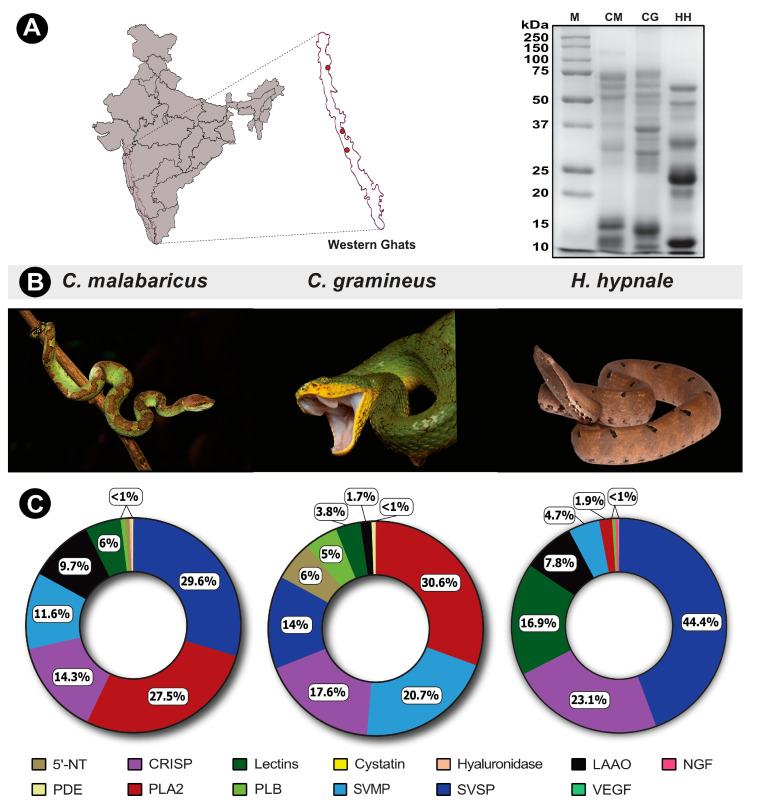
Sampling locations of pit vipers from the Western Ghats, along with their representative photographs and SDS-PAGE and venom profiles. In this figure, panel (**A**) shows sampling locations and SDS-PAGE profiles of pit viper venoms; panel (**B**) shows representative photographs of pit vipers; and panel (**C**) represents the doughnut charts with the relative abundance of toxins in their venoms estimated via tandem mass spectrometry. Toxin families are individually colour coded, and the relative abundances are expressed as percentages. CM: *C. malabaricus* (Photo by Surya Narayanan); CG: *C. gramineus* (Photo by Kartik Sunagar) and HH: *H. hypnale* (Photo by Ajinkya Unawane).

**Figure 2 ijms-24-09516-f002:**
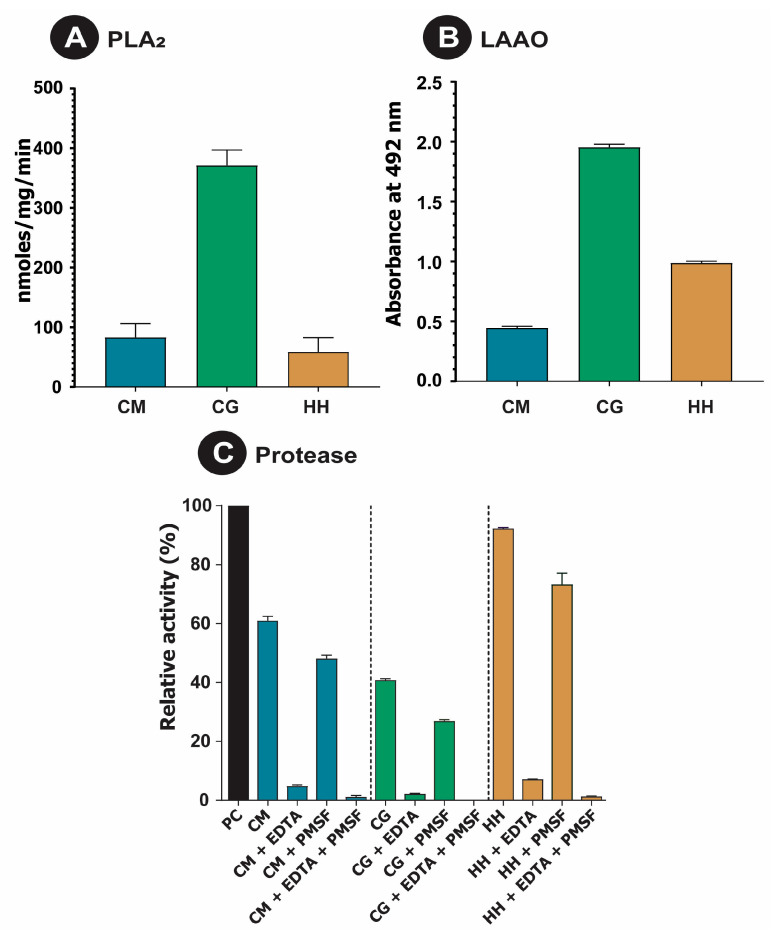
Biochemical activities of pit viper venoms. This figure shows (**A**) PLA_2_, (**B**) LAAO and (**C**) proteolytic (with and without SVMP and/or SVSP inhibitors) activities of *C. malabaricus* (CM), *C. gramineus* (CG), and *H. hypnale* (HH).

**Figure 3 ijms-24-09516-f003:**
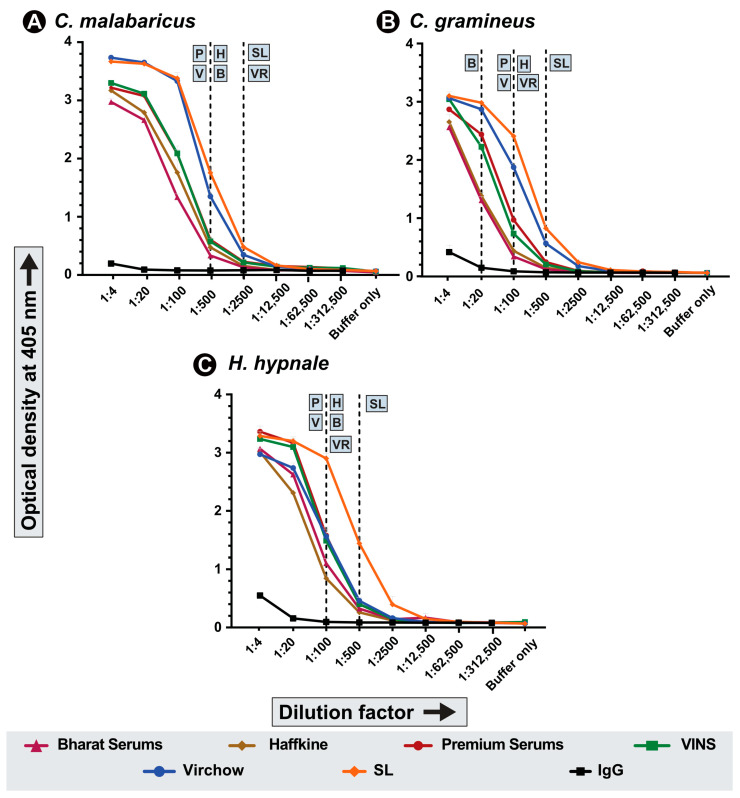
In vitro venom binding of commercial Indian and Sri Lankan antivenoms. Line graphs indicate the in vitro binding potentials of the Indian and Sri Lankan polyvalent antivenoms against the venoms of three pit viper species from the Western Ghats: (**A**) *C. malabaricus*, (**B**) *C. gramineus* and (**C**) *H. hypnale*. The binding abilities were assessed across various dilutions of the antivenom through indirect ELISA experiments. Here, the x- and y-axis represent antivenom dilution and absorbance at 405 nm, respectively. The dotted lines indicate the titre value of the corresponding antivenom: B: Bharat Serums and Vaccines Ltd.; H: Haffkine Bio-Pharmaceutical Corporation Ltd.; P: Premium Serums and Vaccines Pvt. Ltd.; V: VINS Bioproducts Ltd.; VR: Virchow Biotech Private Ltd. and SL: Pentavalent Sri-Lankan antivenom manufactured by Premium Serums and Vaccines Pvt. Ltd.

**Figure 4 ijms-24-09516-f004:**
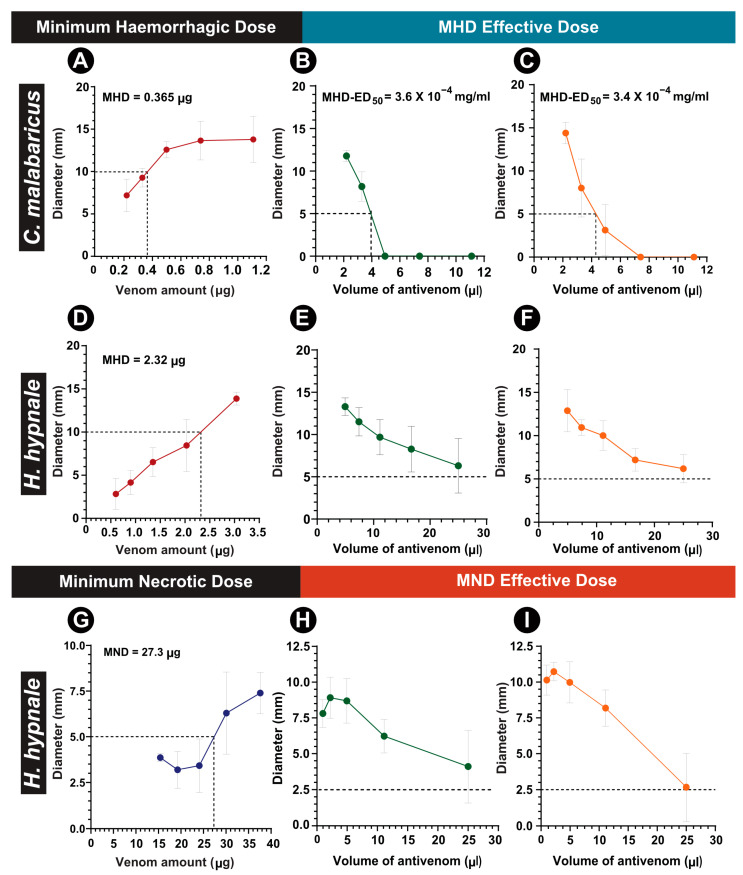
Morbidity assessment of pit viper venoms and its neutralisation by commercial antivenoms. (**A**) MHD of *C. malabaricus* venom, and the (**B**) MHD-ED_50_ of Premium Serum and (**C**) Sri Lankan antivenoms against it. (**D**) MHD of *H. hypnale* venom, (**E**) MHD-ED_50_ of the Premium Serum and (**F**) Sri Lankan antivenom against it. (**G**) MND of *H. hypnale* venom, and (**H**) MND-ED_50_ of Premium Serum, and (**I**) Sri Lankan antivenoms against it.

**Figure 5 ijms-24-09516-f005:**
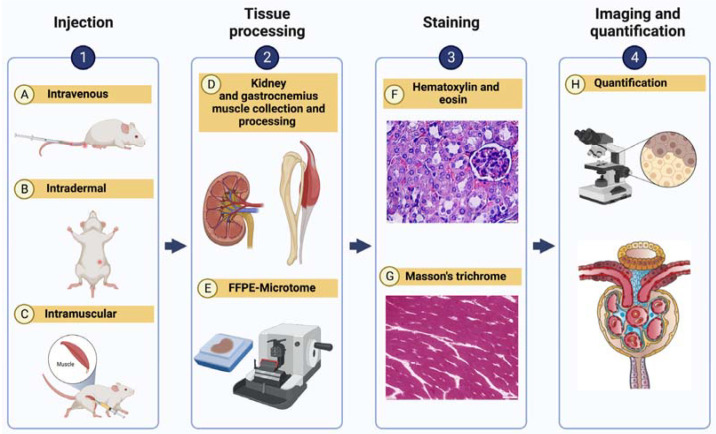
Graphical representation of tissue processing and quantitative microanatomy. Panel (**1**) shows the route of venom administration: (**A**) intravenous, intradermal (**B**), and intramuscular (**C**). Panel (**2**) shows tissue collecting and processing: kidney and gastrocnemius muscle (**D**), followed by (**E**) microtome sectioning. Panel (**3**) shows the dyes used for staining the kidney ((**F**) haematoxylin and eosin) and gastrocnemius muscle ((**G**) Masson’s trichrome). Panel (**4**): (**H**) Quantification of tissue injury was performed via microscopic analyses. Created with BioRender.com.

**Figure 6 ijms-24-09516-f006:**
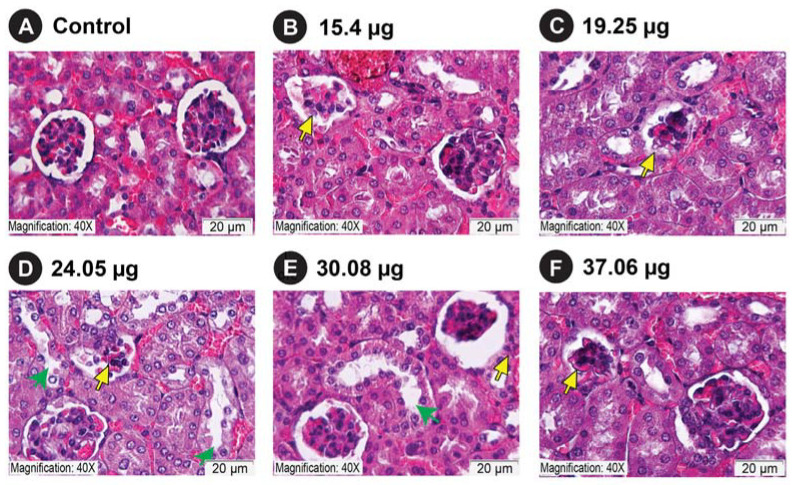
Nephrotoxicity of *H. hypnale* venom from the Western Ghats. Here, panel (**A**) shows control mice and panels (**B**–**F**) represent treatment groups: 15.4 μg (**B**), 19.25 μg (**C**), 24.05 μg (**D**), 30.08 μg (**E**), and 37.06 μg (**F**) concentrations of *H. hypnale* venoms injected intradermally. A scale bar of 20 μm is shown, along with yellow and green arrows that indicate glomerular degeneration and tubular injury, respectively.

**Figure 7 ijms-24-09516-f007:**
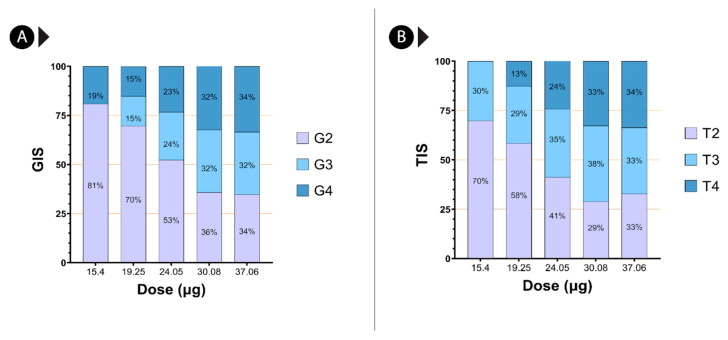
Nephrotic injury score of *H. hypnale* venom from the Western Ghats. Panel (**A**) of this figure highlights GIS, while panel (**B**) highlights the TIS of *H. hypnale* venom.

**Table 1 ijms-24-09516-t001:** Lethal and morbid effects of pit viper venoms and their neutralisation by Indian and Sri Lankan antivenoms. In this table, the LD_50_ for *H. hypnale*, along with the range-finding values of LD_50_ for *C. malabaricus* and *C. gramineus*, are indicated in mg/kg. The estimates of MHD (μg/mouse) and MND (μg/mouse) for *H. hypnale* and/or *C. malabaricus* are also indicated. The amount of undiluted antivenom required to neutralise the lethal and morbid effects is indicated. NA—Not available; Value could not be estimated given insufficient neutralisation; PSVPL (Premium Serums and Vaccines Pvt Ltd.)-IN (Indian) and SL (Sri Lankan).

		C. malabaricus	C. gramineus	H. hypnale
LD_50_ mg/kg		>6	4–6	1.32
MHD (μg/mouse)		0.365	NA	2.32
MND (μg/mouse)		NA	NA	22.3
ED_50_ (mL)	PSVPL-IN	NA	NA	—
	PSVPL-SL	NA	NA	—
MHD_50_ (mL)	PSVPL-IN	3.6 × 10^−4^	NA	—
	PSVPL-SL	3.4 × 10^−4^	NA	—
MND_50_ (mL)	PSVPL-IN	NA	NA	—
	PSVPL-SL	NA	NA	—

## Data Availability

The raw proteomics data generated for this study can be found in PRIDE Database (Accession No: PXD039139).

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
