# Peer review of "Fangs in the Ghats: Preclinical Insights into the Medical Importance of Pit Vipers from the Western Ghats"

_ijms, 2023, doi:10.3390/ijms24119516_

Round 1

Reviewer 1 Report

Very interesting article illustrating the composition and properties of venom from lesser known species. I personally like such articles very much. I like the layout of the article and its content very much and have no serious objections, however, I have a few comments and questions.
1. Please remember to write all Latin names in italics (example lines: 30, 31, 147, 194, 206, 585, 705)
2. I suggest reformatting image 1 and swapping the order of results in section B and C. The whole thing will be more readable if in section B and C the results are in the same order as on the gel in section A. This will make it much easier for the reader to perceive. I also suggest on all other images to stick to this order.
3. Image 1 section C. Are the results shown as doughnut charts from densitometric analysis of the gels or from MS analysis?
4. If there are no restrictions from the editor and publisher, I would like all elements of figure S1 in the main text.
5. Please explain what titre value is, how it was determined and what it really means.
6. Please specify what sex and age the specimens studied were.
7. Please state what concentration of azocasein was used in the test for protease activity.

Author Response

Very interesting article illustrating the composition and properties of venom from lesser known species. I personally like such articles very much. I like the layout of the article and its content very much and have no serious objections, however, I have a few comments and questions.

We are thankful to the reviewer for appreciating the work and for providing important input.

  1. Please remember to write all Latin names in italics (example lines: 30, 31, 147, 194, 206, 585, 705)

Thanks for bringing this to our attention. The version of the manuscript submitted by us had all the above-mentioned Latin names in italics. However, formatting issues cropped up in the version prepared by the journal.

  1. I suggest reformatting image 1 and swapping the order of results in section B and C. The whole thing will be more readable if in section B and C the results are in the same order as on the gel in section A. This will make it much easier for the reader to perceive. I also suggest all other images to stick to this order.

All images, including panels in Figure 1, have been modified as per the reviewer’s suggestion.

  1. Image 1 section C. Are the results shown as doughnut charts from densitometric analysis of the gels or from MS analysis?

The piecharts in Figure 1C represent the relative proteomic abundances of toxin families estimated through tandem MS/MS analysis and normalised with the respective SDS-PAGE band intensities. This has now been clarified in the manuscript as well.

  1. If there are no restrictions from the editor and publisher, I would like all elements of figure S1 in the main text.

Figure S1 has been moved to the main text as Figure 2.

  1. Please explain what titre value is, how it was determined and what it really means.

Purified equine IgG (Bio-Rad Laboratories, USA) was used as the negative control, and absorbance values above the cut-off, calculated as the mean absorbance of the negative control plus two times the standard deviation, were considered for calculating the titre (Senji Laxme et al., 2019; Casewell et al., 2010). The titre value is the first dilution of the antivenom at which the mean absorbance value of the test samples is above the cut-off value. This information has now been added to the manuscript in Line no. 578.

  1. Please specify what sex and age the specimens studied were.

Venoms have been collected only from adult snakes, however, given the forest department restrictions, we were unable to determine the gender of the individuals. This information has now been added to the manuscript.

  1. Please state what concentration of azocasein was used in the test for protease activity.

400 µg of the substrate was used in the protease assay. This value has now been included in the manuscript in Line no. 534.

Reviewer 2 Report

General Impression:

Well conducted and presented research! I have only some minor topics and comments.

Minor topics:

-              species name should always be written in italics (e.g. lanes 30, 31, 147, 206, 374, 378, 422, 585 and 705)

-              some typos: lane 471 "37 °C"; lane 517 "4 M"; lane 539 "0.1 M": lane 540 "0.04 M"

-              the Reference list must be harmonized in style (e.g. journal names, page numbers)

-              Figure 6A: A dose of 15.4 µg results in Nephrotic injury scores of G2 and G4, but not in G3. This is somehow strange. Is it possible that "G4" is "G3" instead?

-              I have problems to understand how the activity factors in lanes 375ff were calculated, e.g. "the haemorrhagic activity of C. malabaricus venom tested here (MHD=0.365 μg/mouse) was 11 times that of H. hypnale (MHD=2.32 μg/mouse)." 2.32 devided by 0.365 results in approximately 6.5, but not 11. Please explain how the factors were calculated.

Comment:

-              I strongly recommend to discuss the "intraspecific variations in the composition of H. hypnale venoms" (lane 322-348) also in an ecological and evolutionary context, e.g. the putative local adaptation of the snake populations to different prey/diet species.

Author Response

General Impression:

Well conducted and presented research! I have only some minor topics and comments.

We are thankful to the reviewer for their kind words and valuable inputs.

Minor topics:

Species name should always be written in italics (e.g. lanes 30, 31, 147, 206, 374, 378, 422, 585 and 705)

Thanks for bringing this to our attention. The version of the manuscript submitted by us had all the above-mentioned Latin names in italics. However, formatting issues cropped up in the version prepared by the journal.

some typos: lane 471 "37 °C"; lane 517 "4 M"; lane 539 "0.1 M": lane 540 "0.04 M"

Thanks for bringing this to our attention. We have incorporated the suggested corrections in the revised version of the manuscript.  

the Reference list must be harmonized in style (e.g. journal names, page numbers)

We have incorporated the suggested correction in the revised version of the manuscript.

Figure 6A: A dose of 15.4 µg results in Nephrotic injury scores of G2 and G4, but not in G3. This is somehow strange. Is it possible that "G4" is "G3" instead?

The stages mentioned in the figure are accurate. We observe either the damage to be too high (G4) or low (G2). Intermediate (G3) stage of damage was not observed. We believe that at doses such as 15.4 µg, there might be predominantly low damage (G2) to most of the regions, with the exception of certain areas experiencing high damage (G4) initially.

I have problems to understand how the activity factors in lanes 375ff were calculated, e.g. "The haemorrhagic activity of C. malabaricus venom tested here (MHD=0.365 μg/mouse) was 11 times that of H. hypnale (MHD=2.32 μg/mouse)." 2.32 divided by 0.365 results in approximately 6.5, but not 11. Please explain how the factors were calculated.

Thanks for bringing this to our attention. Indeed, this was a mistake from our end. It is, indeed, 6.5 times and not 11. This has now been corrected in the manuscript.

I strongly recommend to discuss the "intraspecific variations in the composition of H. hypnale venoms" (lane 322-348) also in an ecological and evolutionary context, e.g. the putative local adaptation of the snake populations to different prey/diet species.

Since we do not know what factors are actually responsible for the observed venom variation, we refrained from discussing this in the previous version of the manuscript. However, following the reviewer’s suggestion, we have now added a few lines discussing the influence of the ecology and environment on venom variation.

Reviewer 3 Report

The paper reports a well-designed research, combining biochemical, hystological and toxicological approaches. The only remark is about usage of one-domension electrophoresis for proteomics. Usually 2D electrophoresis is used for this purpose. The authors should provide evidence, that 1D electrophoresis was sufficiently potent for their work.  

Author Response

The paper reports a well-designed research, combining biochemical, histological and toxicological approaches. The only remark is about usage of one-domension electrophoresis for proteomics. Usually 2D electrophoresis is used for this purpose. The authors should provide evidence, that 1D electrophoresis was sufficiently potent for their work.

Mass spectrometry followed by 1D SDS-PAGE is a strategy predominantly used in the field of venomics (Kalita et al., 2018, Vanuopadath et al., 2018, Vanuopadath et al, 2017, Choudhury et al., 2017). In contrast, 2D electrophoresis is not a commonly adapted methodology. Moreover, the highly sensitive mass-spectrometry technique adopted in this paper has provided sufficient resolution and has even identified several minor venom components, including hyaluronidases, nerve growth factors and phosphodiesterases.